# NOTO Transcription Factor Directs Human Induced Pluripotent Stem Cell-Derived Mesendoderm Progenitors to a Notochordal Fate

**DOI:** 10.3390/cells9020509

**Published:** 2020-02-24

**Authors:** Pauline Colombier, Boris Halgand, Claire Chédeville, Caroline Chariau, Valentin François-Campion, Stéphanie Kilens, Nicolas Vedrenne, Johann Clouet, Laurent David, Jérôme Guicheux, Anne Camus

**Affiliations:** 1INSERM UMR 1229, RMeS, Université de Nantes, ONIRIS, F-44042 Nantes, FranceNicolas.Vedrenne@univ-nantes.fr (N.V.);; 2CHU Nantes, PHU 4 OTONN, F-44042 Nantes, France; 3Nantes Université, CHU Nantes, INSERM, CNRS, SFR Santé, FED 4203, Inserm UMS 016, CNRS UMS 3556, F-44042 Nantes, France; 4Nantes Université, CHU Nantes, INSERM, CRTI, UMR 1064, ITUN, F-44042 Nantes, France; 5CHU Nantes, Pharmacie Centrale, PHU 11, F-44042 Nantes, France

**Keywords:** human induced pluripotent stem cells, intervertebral disc regeneration, mesendoderm progenitors, notochord, directed differentiation, signalling, stem cell therapy

## Abstract

The founder cells of the Nucleus pulposus, the centre of the intervertebral disc, originate in the embryonic notochord. After birth, mature notochordal cells (NC) are identified as key regulators of disc homeostasis. Better understanding of their biology has great potential in delaying the onset of disc degeneration or as a regenerative-cell source for disc repair. Using human pluripotent stem cells, we developed a two-step method to generate a stable NC-like population with a distinct molecular signature. Time-course analysis of lineage-specific markers shows that WNT pathway activation and transfection of the notochord-related transcription factor NOTO are sufficient to induce high levels of mesendoderm progenitors and favour their commitment toward the notochordal lineage instead of paraxial and lateral mesodermal or endodermal lineages. This study results in the identification of NOTO-regulated genes including some that are found expressed in human healthy disc tissue and highlights NOTO function in coordinating the gene network to human notochord differentiation.

## 1. Introduction

The intervertebral disc (IVD) is a fibrocartilaginous joint composed of a hydrated gel-like central part, the nucleus pulposus (NP), where large vacuolated notochordal cells (NC) and chondrocyte-like cells (CLC) reside [1,2]. The mature NC population has been well identified as a key regulator of disc homeostasis [3,4,5]. Indeed, around the age of skeletal maturity, the loss of NC followed by the decline of CLC viability is a primary event leading to degenerative disc disease (DDD) [6]. This condition results in impaired biomechanical functions of the IVD and causes low back pain. Current treatment strategies focus on pain management or surgical intervention with limited efficacy. The lack of disease-modifying therapeutics for DDD is linked to our limited understanding of the cellular and molecular mechanisms that regulate IVD development, maturation and health. In the past decade, studies have demonstrated that NC-secreted factors exert significant regenerative effects on human CLC derived from degenerated IVD by promoting anabolic activities and NP-like matrix synthesis [7,8,9,10,11]. The limited availability of resident NC cells and the difficulty of maintaining them in vitro make the use of pluripotent stem cells (PSC) an attractive alternative cell-based strategy for disc repair [12,13]. Lineage tracing studies in mouse model demonstrated that the founder cells of the NP originate in the embryonic notochord [14,15]. The notochord, or axial mesoderm, is a transient signalling structure involved in the regionalization of adjacent embryonic tissues such as the neural tube, the gut and its derivatives, and the paraxial mesoderm forming the somites [16]. Through secretion of SHH and NOGGIN, the notochord induces the migration and differentiation of the somite-derived sclerotomal cells to form the spine with alternating vertebrae and annulus fibrosus (IVD fibrous peripheral part). During organogenesis, NC accumulate and persist solely in the presumptive intervertebral region to form the NP [17,18].

Only one study has reported mouse embryonic stem cell differentiation into notochord-like cells (NLC) [19]. Several previous studies reported the production of NP-like cells by directing human induced PSC (hiPSCs) toward notochordal differentiation [20,21]. However, a thorough lineage monitoring will be required to ascertain the hiPSCs commitment to axial mesoderm/notochord lineage, rather than to the paraxial mesoderm/somitic lineage at the onset of musculoskeletal tissues formation [22,23]. Such a limitation highlights the need for optimized methodologies to generate human NLC for further investigations of its biology and for future potential clinical translation.

Fate-mapping, genetic and transcriptomic studies have demonstrated that axial mesoderm is distinct from the pan-mesoderm as it descends from the node/organizer region and expresses specific sets of genes [24,25,26,27]. In the mouse embryo, the onset of notochordal lineage is controlled by the transcription factors *Brachyury* (*T*) and *Foxa2.* Both factors are required for the expression of the notochordal transcription factor *Noto* [28,29,30]. Although the invalidation of the *Noto* gene results in moderate defects in node and posterior notochord formation, cell-tracking study in the mouse embryo demonstrates its pivotal role in the maintenance of notochordal identity [31,32]. Indeed, in the absence of *NOTO*, mutant axial mesoderm cells are found mis-located in the paraxial/somitic mesoderm and downregulate *T* and *Foxa2* expression [33]. In human, *NOTO* expression pattern and function has not been elucidated [34].

Basic knowledge from the mouse model was used as a general framework in this study to investigate how WNT, ACTIVIN/NODAL, FGF and SHH signalling pathways drive hiPSCs differentiation into the notochordal lineage. Developmental paths and differentiation outcomes (endoderm, paraxial and lateral mesoderm, and axial mesoderm/notochord lineages) were characterized at RNA and protein levels using lineage specific markers. By providing *NOTO* mRNA, we demonstrated that hiPSCs differentiate towards a phenotypically stable NLC population, and remarkably express markers found in human healthy disc tissue. This study reports the identification of the whole transcriptomic signature of human NLC.

## 2. Materials and Methods

### 2.1. Reprogramming, Validation and Culture of Human Induced Pluripotent Stem Cells

Human iPSCs were generated from dermal fibroblasts and had normal karyotypes, no gain of SNP compared to parental fibroblasts. Pluripotency was assessed by teratoma formation and trilineage differentiation [35]. Human iPSCs lines used in this study were LON71-002, LON71-019 and PB174-005 and were maintained on matrigel-coated plates with mTeSR1 medium from 25 up to 40 passages. Gentle TryplE enzymatic digestion was performed twice a week for hiPSCs expansion.

### 2.2. Differentiation of Human Induced Pluripotent Stem Cells

For differentiation, hiPSCs were stimulated with CHIR99021 (CHIR) and/or Activin A (ActA) in a N2B27 medium. After 2 days of stimulation, cells were transfected for 3 consecutive days with synthetic mRNA encoding for T, FOXA2 or NOTO. Differentiated cells were maintained in N2B27 supplemented with CHIR, and FGF2 or SHH factors. Detailed experimental procedures and the list of reagents are provided in Figure 1 and Appendix A (List of reagents used for hiPSCs culture and differentiation).

### 2.3. RNA Extraction and RT-qPCR

One microgram of total RNA extracted with the Nucleospin II RNA Kit (740955, Macherey Nagel) was reverse transcribed using SuperScript III First Strand synthesis kit (11752, Life technologies, Carlsbad, CA, USA). Quantitative RT-PCR experiments were performed using TaqMan technology and fold change represented using a base 2 logarithm determined by the Livak Method (Relative quantification RQ = 2^−ΔΔCq) [36]. Endogenous *T*, *FOXA2* and *NOTO* transcripts were measured by SybR green technology. Taqman and primers used are listed in Appendix A (List of Taqman Assays and Primer sequences for RT-qPCR analysis by SYBR GREEN technology).

### 2.4. Immunostainings

Cells were fixed with 4% paraformaldehyde for 15 min, following by a permeabilization step and then blocked in 3% bovine serum albumin for 30 min. Immunostaining conditions for FOXA2, T, SOX9 and SOX17 are detailed in Appendix A (Antibodies and dilutions used for Immunofluorescence experiments). Nuclei were then counterstained with Hoechst (H3569, Life technologies, Carlsbad, CA, USA) before imaging with a confocal microscope A1Rsi (Nikon, Champigny-sur-Marne, France). The percentage of T+/FOXA2+, T+/SOX9+ or FOXA2+/SOX17+ double positive cells was defined using Volocity^®^ software version 6.0.0.

### 2.5. cDNA Libraries, 3′ Digital Gene Expression RNA-Sequencing (DGE-seq), and Bioinformatic Analyses

To generate 3′-DGE libraries, Poly(A)+ mRNA were converted to cDNA decorated with universal adapters, sample-specific barcodes and unique molecular identifiers (UMIs) using a template-switching reverse transcriptase [37]. Differential expression analysis has been performed using DESeq2 in R (https://doi.org/10.1186/s13059-014-0550-8). Hierarchical clustering heatmaps were generated by complex heatMaps package in R (https://doi.org/10.1093/bioinformatics/btw313). Our transcriptomic data were compared to Tsankov et al., datasets (NCBI: GSE17312). Gene Ontology enrichment analyses were performed using Panther database. The raw read sequence data and sample annotations generated in this study are available at the European Nucleotide Archive (ENA) with the accession number PRJEB18663.

### 2.6. Statistical Analysis

Data are representative of the number of independent experiments as indicated in the figure legend. Mean values ± SEM are calculated when possible. As a consequence of the high number of experimental conditions and time points, no adequate statistical analysis can be provided, except for experiments in Figure 4C. Statistical analysis (2 way Anova test) is shown in Appendix C.

## 3. Results

### 3.1. WNT Activity Induces High Levels of Mesendoderm Progenitors

WNT/β-CATENIN and NODAL/SMAD2/3 signalling are the two main pathways used in vitro to model the induction of the primitive streak (PS) and mimic early developmental events leading the formation of mesoderm and endoderm germ layers [38,39]. We first intended to decipher the contribution of those pathways during hiPSCs differentiation (see schematic workflow of hiPSCs differentiation, Figure 1).

Human iPSCs cultured for 2 days with 3 or 6 µM CHIR (activation of canonical WNT/β-CATENIN signalling via a selective small molecule inhibitor of GSK3) exhibited low level of pluripotency markers, high level of *LEF1*, *NODAL* and *LEFTY1* transcripts and changes in cell morphology, indicating cell differentiation (Figure 2A–C). No sign of differentiation was observed at 1µM CHIR treatment. Interestingly, hiPSCs stimulated with 3 µM CHIR expressed the PS markers *BRACHYURY* (*T*), *MIXL1* and *EOMES* together with high expression of the Anterior PS (APS) markers *FOXA2*, *GSC* and *CER1* [40]. In contrast, when hiPSCs were treated with 6 µM CHIR, cells acquired a Posterior PS-like (PPS-like) identity revealed by elevated level of *T* and *MIXL1* transcripts and conversely lower levels of *NODAL*, *EOMES*, *FOXA2* and *CER1* [41]. Both conditions resulted in mostly T immunopositive cells indicating an early PS-like identity at day 1 (Figure 2D). At day 2, 3 µM CHIR treatment induced 88% ± 5.5% of T+/FOXA2+ immunostained mesendoderm progenitors whereas 6 µM CHIR triggered commitment towards mesoderm progenitors with 95% ± 2.5% of T+/FOXA2- cells (Figure 2D).

We next investigated the differentiation outcome of hiPSCs treated with 3 µM CHIR supplemented with increasing doses of Activin A (ActA) (Figure 3A). No change in expression of *LEF1* and *NODAL* was observed at day 1. At day 2, the down-regulation of *NODAL* and *LEFTY* expression indicated the activation of the negative feedback loop of NODAL/SMAD2/3 signalling (Figure 3B) [42,43]. *FOXA2* and *CER1* expression increased upon treatment with 10 ng/mL ActA, consistent with the role of ACTIVIN/NODAL in endoderm specification. No apparent effect in pluripotency status or cell morphology occurred with the addition of ActA (Figure 3B,C). Remarkably, while cells were predominantly positive for T immunostaining at day 1 (Figure 3D), T+/FOXA2+ mesendoderm progenitor cells was reduced from 65% ± 1.5% to 53% ± 8% with increasing doses of ActA at day 2 (Figure 3D). Altogether, these results showed that CHIR acts as potent inducer of mesendoderm progenitors in the hiPSCs model. The addition of ActA in the differentiation medium led to a decrease of these progenitors.

### 3.2. WNT and ACTIVIN/NODAL Activities Induces Mesendoderm Progenitors with Distinct Lineage Competencies

To evaluate the ability of hiPSC-derived mesendoderm progenitor cells (MEPC) to generate notochordal lineage, we treated hiPSCs with 3 µM CHIR, stimulated 1 or 2 days with or without 2 ng/mL of ActA (Figure 4A). Immunostaining quantification revealed that the optimal strategy was 3 µM CHIR stimulation for 2 days (88% ± 5.5% MEPC) compared to 3 µM CHIR + 2 ng/mL ActA for 1 or 2 days (12% ± 1% and 65% ± 1.5% MEPC, respectively; Figure 4B). Cells were further cultured for 3 days with sustained 3 or 6 µM CHIR, to mimic the function of WNT signalling in the maintenance of notochordal fate during mouse axis elongation [44,45,46]. Gene expression analysis showed that MEPC sustained with 6 µM CHIR differentiated towards mesoderm lineages as demonstrated by higher expression of *MIXL1*, *TBX6* and *FOXF1* (Figure 4C). In contrast, *FOXA2*, *T*, *SHH*, *FOXJ1* and *NOGGIN* transcripts were detected when MEPC were cultured further with 3 µM CHIR. Interestingly, *FOXA2*, *T*, *SHH*, *FOXJ1* and *NOGGIN* transcripts were also detected when cells were co-stimulated with CHIR and ActA. However, ActA supplementation at the beginning of the differentiation protocol correlated with greater endoderm specification as shown by increased *FOXA2* and *SOX17* expression. Altogether, these results refine our understanding of the respective influence of both WNT and ACTIVIN/NODAL signalling on lineage specification: ACTIVIN/NODAL activity directs MEPC differentiation towards endoderm fate rather than mesoderm fate, while mesoderm fate is promoted by high WNT activity.

In mouse, *NOTO* gene expression delineates organizer regions where axial mesoderm progenitors are found. Later, *Noto* marks node-derived posterior axial mesoderm/notochord until early organogenesis. Remarkably, the absence of axial mesoderm/notochord progenitors and progenies at day 5, as demonstrated by the lack of *NOTO* expression and T+/FOXA2+ cells (data not shown), indicates that MEPC did not differentiate toward notochord lineage in any condition analysed. This result suggests that neither the combined activation of both pathways, nor the continuous activation of WNT pathway, is sufficient to sustain notochordal fate. Based on these findings, and to circumvent definitive endoderm differentiation, hiPSCs were treated with 3 µM CHIR to generate high levels of MEPCs for the remainder of the study.

### 3.3. NOTO Transcription Factor Triggers MEPC Commitment toward Notochordal Fate

The early loss of T+/FOXA2+ cells during the course of hiPSCs differentiation argues against the presence of NLC. We thus investigated whether forced expression of *T*, *FOXA2* or *NOTO* factors could trigger the commitment of MEPC toward notochordal fate. Synthetic mRNAs encoding for T, FOXA2 and NOTO were independently transfected daily from day 2 to day 4 in MEPC maintained in 3 µM CHIR (Figure 5A). The time-course analysis of lineage specific markers revealed three distinct differentiation outcomes, with *T* transfection leading to an increase in paraxial and lateral mesoderm markers (*MIXL1*, *TBX6* and *FOXF1*), while *FOXA2* or *NOTO* transfection resulted in a significant increase in axial mesoderm markers (*T*, *FOXA2*, *NOTO*, and *SHH*; Figure 5B). The presence of T+/FOXA2+ cells at day 7, when *NOTO* was transfected only, confirmed the presence of NLC (6.6%, Figure 5C,D and Appendix A—Immunostaining at day 3 and day 5 in *T*-, *FOXA2*-, and *NOTO*- transfected cells). In the course of the notochordal maturation process, we expected based on the mouse embryonic studies, a down-regulation of immature markers *FOXA2* and *NOTO* and conversely, an up-regulation of the transcription factors *SOX-5*, *-6* and *-9* as a consequence of the activation of SHH signalling [47,48,49,50]. These *SOX-* gene markers are detected both in NC and somite-derived sclerotomal cells. In order to discriminate between these two cell-types, we performed co-immunostaining analysis. The results confirmed the presence of T+/SOX9+ NLC up to day 7 when *NOTO* was transfected (7.6%, Figure 5C,D and Appendix A). In the course of hiPSCs differentiation, FOXA2+ is also indicative of the presence of nascent definitive endoderm cells, which co-express SOX17 at early stages [51,52]. Consistent with our RT-qPCR results, *FOXA2*-transfected cells had a propensity to differentiate into definitive endoderm cells as compared to those transfected with *NOTO* (45.6% and 26.2% of FOXA2+/SOX17+ cells at day 7 respectively; Figure 5C,D and Appendix A). Altogether the results support the hypothesis that amongst all three transcription factors transfected, *NOTO* directs the commitment of MEPC toward notochordal lineage.

### 3.4. NOTO mRNA Transfection and WNT Signalling Activity Are Sufficient to Induce a Stable Notochord Population

In mice, FGF and SHH activities are required for the emergence of different mesoderm subtypes and in the maintenance of the notochordal lineage during embryonic axis elongation [53,54]. Thus, we sought to test the effect of exogenous FGF or SHH on the proportion of stable NLC in the hiPSCs differentiation model. When MEPC were transfected with *NOTO* mRNA in the presence of 50 ng/mL FGF2 from day 2 to day 5, notochordal markers remained unchanged, except for a slight up-regulation of endogenous *T* and *NOTO*, and the proportion of FOXA2+/T+ cells was maintained (Figure 6A–C). In contrast, FGF2 supplementation led to a significant increase in paraxial and lateral mesoderm markers.

Lastly, we investigated the differentiation outcome when FGF2 or SHH was supplemented during the last phase of differentiation (Figure 7A). *NOTO* mRNA transfection induced high expression of notochord-related markers between day 3 and day 5 and maintenance until day 7, while the expression the endoderm marker *SOX17* was down-regulated with time (Figure 7B). Only a slight decrease in *T* and *SHH* expression was noticeable from day 5 when FGF2 was added. A similar proportion of T+/FOXA2+ cells were observed in all the *NOTO*-transfected conditions, whether complemented or not with FGF2 or SHH (Figure 7C). This result indicates that addition of exogenous FGF2 and SHH ligands did not enhance, neither the differentiation of MEPC into NLC nor their maintenance in vitro. *NOTO* mRNA transfection and sustained WNT signalling activity are sufficient to induce a stable NLC population.

Broader gene expression was analysed to characterize the nature of NLC emerging when *NOTO* mRNA is transfected. The expression of FGF and SHH pathway target genes *SPRY1* and *GLI1* respectively were found strongly up-regulated from day 5 in *NOTO*-transfected cells (Figure 7D). This could account for the small changes observed above in the general expression profile following FGF2 and SHH supplementation. The nascent-mesoderm marker *CDH2* and notochordal markers *FOXA1* and *FN1* were also induced and maintained up to day 7 in MEPC when transfected with *NOTO*, suggesting similarities between notochordal differentiation in vitro and mouse notochord development. *SOX5*, *SOX6* and *SOX9* genes are important regulators of NC survival and chondrogenesis in IVD development by controlling the synthesis of common extracellular matrix component such as *AGGRECAN* and *TYPE II COLLAGEN* [47,48]. It should be noted that these markers were detected at relatively higher levels in control compared to *NOTO*-transfected condition. This result indicated that somite-derived sclerotomal cells differentiation is major in untransfected control. Although *KRT18* and *CDH2* were strongly induced, cytosolic vacuolar structures typically found in human mature juvenile NC were not observed [55,56,57,58]. In addition, the low expression of *CA12* and *LGALS3* at day 7 indicated that differentiated NLC maintains an immature/embryonic state.

### 3.5. Molecular Characterization of NOTO- and FOXA2-Directed MEPC Differentiation

RNAseq analysis was performed in order to understand molecular events arising during differentiation following *NOTO* mRNA transfection in the presence of 3 µM CHIR (Figure 8A). We first assessed expression levels of mesoderm, ectoderm and endoderm markers previously characterised by Tsankov et al. in control, *NOTO*- and *FOXA2*- transfected cells [59]. This integrative analysis revealed that the control most resembled mesoderm, *FOXA2*-transfected condition resembled endoderm but *NOTO*-transfected cells displayed distinct transcriptomic signature (Figure 8B). Genes differentially expressed between the three conditions: control, *NOTO*- and *FOXA2*- transfected cells highlighted five clusters with similar expression trends (Figure 8C): (i) Genes readily induced in *NOTO*-transfected cells, reaching their maximal expression at day 3 and maintained until day 7 (“Immediate NOTO response genes”, shown Figure 8D), (ii) Genes induced in the *NOTO* condition, at day 3 but reaching their maximal expression at day 7 (“Delayed NOTO response genes” shown Figure 8E), (iii) Genes inhibited by *NOTO* (shown in Appendix A; Details of the transcriptomic cluster presented in Figure 8C), (iv) Mesendoderm-related genes (shown in Appendix A; Details of the transcriptomic cluster presented in Figure 8C), and (v) Genes induced in *FOXA2*-transfected cells (“FOXA2 response genes”, shown in Appendix A; Details of the transcriptomic cluster presented in Figure 8C). Intersection of the *NOTO* response gene expression profiles with datasets from Tsankov et al., showed that *NOTO*-transfected cells display a unique signature composed of clusters (i) and (ii), not observed in the three germ layers (Figure 8D,E). Mesendoderm genes were transiently expressed at day 3 in all differentiation conditions, but their prolonged expression was only observed in *NOTO*-transfected cells (Appendix A). Some of these genes were distinctively expressed in mesoderm or endoderm cells, but none in ectoderm cells (Appendix A). Conversely, genes specifically inhibited by *NOTO* are not expressed in mesendoderm at day 2 but are expressed in one of the three germ layers (Appendix A), suggesting that *NOTO* also blocked the commitment toward other germ layers. Our transcriptomic analysis indicates that the overexpression of *NOTO* during hiPSCs differentiation while maintaining mesendoderm-related genes also prevents cells from differentiating into mesoderm, ectoderm or endoderm layers. Finally, functional enrichment analysis highlighted “anterior/posterior axis specification” and “notochord development” associated with *NOTO*-transfected cells (Figure 8F), supporting our conclusion that expression of *NOTO* in MEPC induced notochordal fate. Hence, this transcriptomic analysis provides the first molecular signature of hiPSCs-derived NLC.

## 4. Discussion

In humans, limited knowledge is available on signalling pathways and gene network orchestrating the formation of the node and notochord [60,61]. Sequential experiments in the present study demonstrate that the differentiation of hiPSCs towards endoderm and mesoderm lineages is effective by modulating WNT, ACTIVIN, and FGF signalling pathways [62,63]. Our results validate the use of CHIR99021 as a potent inducer of hiPSCs differentiation towards mesendoderm progenitors without commitment toward ectodermal/neuroectodermal lineages. The absence of expression of the cardiac markers HAND1 and HAND2 and the lack of beating cells indicated the failure of hiPSCs to form cardiomyocyte-like cells using this differentiation protocol. We show that the use of intermediate concentration of CHIR promotes mesendoderm progenitors with APS-like identity, which is favourable to the emergence of the notochordal lineage. ActA supplemented CHIR treatment severely reduces the proportion of mesendoderm progenitors and then promotes endodermal fate commitment.

*BRACHYURY* (*T*) is expressed early in PS and nascent endoderm and mesoderm lineages. From early organogenesis, its expression is restricted to axial mesoderm/notochord lineage and further maintained after birth in NC constituting the NP, the core of the IVD, in mouse and in human [64,65]. In a recent report, *T*-encoding plasmid transfection was shown to reprogram mildly degenerate human CLC in vitro to a healthy NP-like phenotype with increased expression of key NP markers and significant proteoglycan/glycosaminoglycan accumulation [66]. Another interesting study also used *T*-encoding plasmid transfection to differentiate hiPSCs toward an NLC phenotype capable of synthesizing a proteoglycan-rich matrix and playing a protective role in the catabolic environment of injury-induced porcine disc model [67]. Previous work has revealed that T genomic targets in differentiating PSC vary based on cellular, developmental and signalling contexts [59,68]. Here, we report that sustained expression of *T* in mesendoderm progenitors was not sufficient for their further differentiation into NLC, despite SHH and FGF2 supplementation.

Our study provides evidence that amongst all three transcription factors required for axial mesoderm development in mouse, *NOTO* triggers the commitment of mesendoderm progenitors toward notochordal fate as demonstrated by the upregulation of notochord-associated markers. This study demonstrates that transient expression of *NOTO* allows mesendoderm progenitors to maintain the lineage-specific expression of the two key notochordal regulatory factors, *FOXA2* and *T* (Figure 9). Our results support the hypothetical model that *NOTO* confers axial mesoderm stability to the promiscuous state of the bipotent mesendoderm progenitors, preventing differentiation towards the mutually exclusive endoderm and mesoderm fates (Figure 9). *NOTO* factor may exert its transcriptional activity via the stabilization of *FOXA2* and *T* transcription complexes required to regulate the molecular program of notochord formation. In mouse, this hypothesis is supported by the existence of putative binding sites for *FoxA2* or *T* in notochordal-related gene promoters and by the model of gene regulatory network of node/notochord proposed by Tamplin et al. [25,28,30]. Whether this model is valid for notochordal lineage commitment in human remains to be proved. This model does not exclude the possibility that *NOTO* interacts with other partners and that this diversity of interaction enables axial mesoderm/notochord fate specification. Remarkably, *NOTO* mRNA transfection in MEPC resulted in a significant increase of endogenous *NOTO* gene expression. Several reports in the literature provided evidence that *Noto* activates or represses its own expression depending on the context. Data from the zebrafish model showed downregulation of *flh* expression (*Noto* homolog) in *flh*^n1^ mutants suggesting that *flh* positively regulates its own expression [69]. In the mouse model, the loss of *Noto* function resulted in persistent *Noto* expression in anterior region of the embryo suggesting that *Noto* is required for its own repression [31]. *FOXA2* mRNA transfection in MEPC also resulted in an increase of endogenous *NOTO* gene expression. Several binding sites for FOXA2 transcription factor are found in the NOCE (Node and nascent notOChord Enhancer) within the ci-regulatory region of the murine *Noto* gene [29]. Although a pivotal role of FOXA2 for the activation of other identified notochord enhancers has been described [28,70], in the case of the *Noto* gene, it is more likely that FOXA2 acts cooperatively with other factors to activate the NOCE enhancer [29]. Note that NOCE contains a HOX binding site suggesting the possibility that NOTO regulates its own transcription. These positive feedback loops involving several transcriptional regulators that reinforce expression of specific lineage markers may participate to the stabilization of the notochordal identity. Whether the regulation of the human NOTO gene is mediated directly by these transcription factors remained to be addressed in order to understand gene regulatory networks that control human notochord development.

Further refinement of the culture conditions is required to exclude the commitment to alternative cell fates. Variation in the differentiation outcome can be explained by cells’ response to endogenous/paracrine signalling. Further investigations of NLC by single cell RNAseq will allow us to decipher specific regulatory networks driving notochord fate specification in humans. In the future, 3D culture will be investigated and ultimately with the use of material with biophysical properties. This may optimize notochordal differentiation efficiency and be supportive of NLC maintenance, particularly with extended culture duration, as well as maturation toward an adult NLC phenotype.

Our method achieves an essential step and lays the groundwork for future studies in generating therapeutically useful hiPSC-derived cells for IVD regeneration. NLC production will allow further study on their biology and NC-associated secreted regulatory molecules to pave the way for the characterization of essential players for healthy disc maintenance. 

## Figures and Tables

**Figure 1 cells-09-00509-f001:**
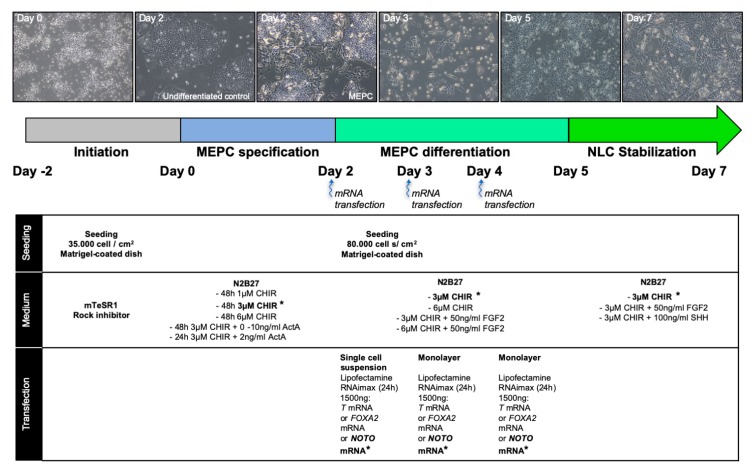
Schematic workflow of hiPSCs differentiation. The differentiation was initiated by single cell seeding at 35.000 cells/cm^2^ (TryplE digestion) on matrigel-coated plates in mTser1 medium supplemented with rock inhibitor for 24 h. From day 0 to day 2, hiPSCs were cultivated in N2B27 in increasing doses of CHIR99021 and Activin A for hiPSC-derived mesendoderm progenitor cell (MEPC) specification. At Day 2, MEPC were dissociated with TryplE and transfected with Lipofectamin RNAimax (5:1) in a single cell suspension with 1500 ng of *T*, *FOXA2* or *NOTO* mRNA for 24 h for MEPC differentiation. Monolayer transfections were then performed on day 3 and day 4. Cells were maintained in N2B27 with 3 or 6 µM CHIR99021 with or without 50 ng/mL FGF2 from day 2 to day 5. For the stabilization phase, transfected cells were maintained in N2B27 supplemented with 3 µM CHIR99021 with or without 50 ng/mL FGF2 and 100 ng/mL SHH from day 5 to day 7. Top panel: representative brightfield images of differentiating hiPSCs upon optimal culture condition for notochordal lineage from day 0 to day 7, including undifferentiated control cells at day 2 (cells without treatment). (*) indicates optimal culture condition for notochordal differentiation at day 7.

**Figure 2 cells-09-00509-f002:**
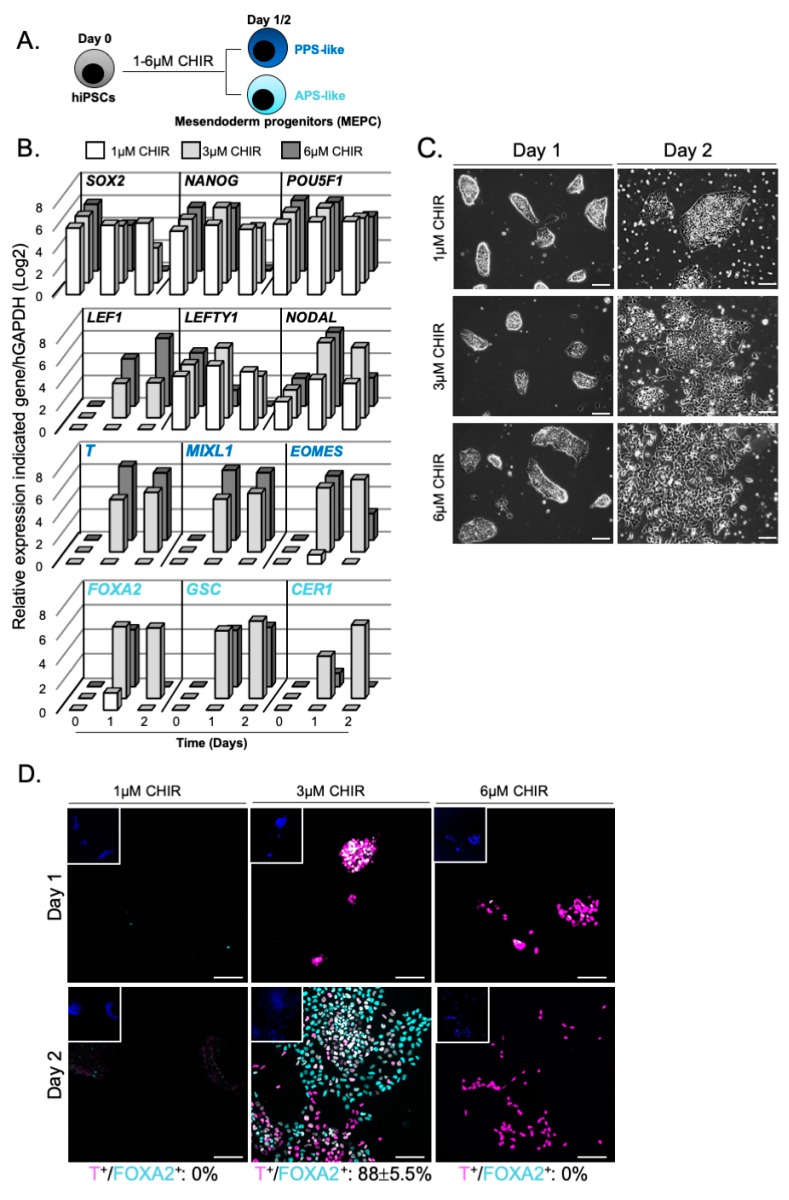
WNT signalling pathway induces hiPSCs differentiation towards mesendoderm progenitors. (**A**) Modulation of WNT signalling by CHIR; (**B**) Relative expression of pluripotent markers (*SOX2*, *NANOG* and *POU5F1*), WNT and NODAL target genes (*LEF1*, *LEFTY1* and *NODAL*), primitive streak (*T*, *MIXL1* and *EOMES*) and mesendoderm markers (*FOXA2*, *GSC* and *CER1*), (n = 2 independent experiments, mean values); (**C**) Brightfield acquisition of differentiating hiPSCs upon CHIR treatment; (**D**) Immunostainings of T+/FOXA2+ positive cells (cell counting at day 2, n = 2 independent experiments, mean percentage ± SEM). Insets are showing nuclei staining with Hoechst. Scale bars: 100 µm. APS = anterior primitive streak; PPS = posterior primitive streak.

**Figure 3 cells-09-00509-f003:**
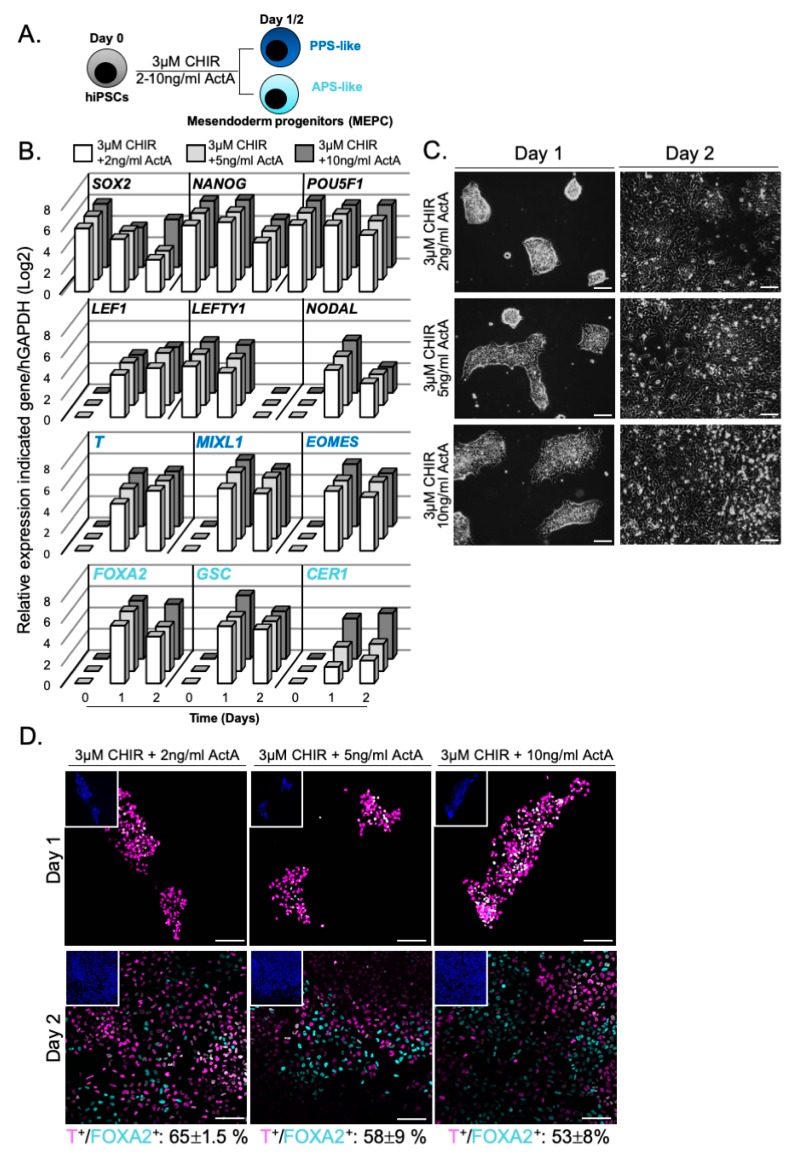
WNT and NODAL signalling pathways induce hiPSCs differentiation towards mesendoderm progenitors. (**A**) Modulation of WNT and NODAL signalling by CHIR and ActA; (**B**) Relative expression of pluripotent markers (*SOX2*, *NANOG* and *POU5F1)*, WNT and NODAL target genes (*LEF1*, *LEFTY1* and *NODAL*), primitive streak (*T*, *MIXL1* and *EOMES*) and mesendoderm markers (*FOXA2*, *GSC* and *CER1*), (n = 3, mean values); (**C**) Brightfield acquisition of differentiating hiPCSs upon CHIR and ActA treatment; (**D**) Immunostainings of T+/FOXA2+ positive cells (cell counting at day 2, n = 2 independent experiments, mean values ± SEM). Insets are showing nuclei staining with Hoechst. Scale bars: 100 µm. APS = anterior primitive streak; PPS = posterior primitive streak.

**Figure 4 cells-09-00509-f004:**
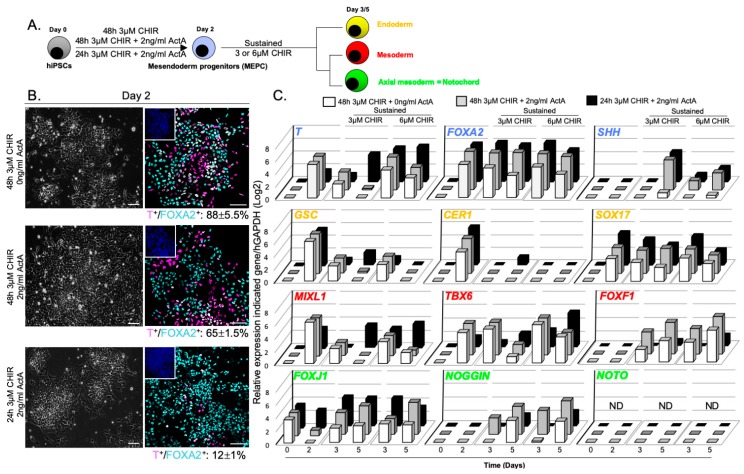
Generation and differentiation of mesendoderm progenitors upon WNT and NODAL signalling activation. (**A**) Effects of sustained WNT signalling activation on the differentiation of MEPC; (**B**) Brightfield acquisition of MEPC and immunostainings of T+/FOXA2+ positive cells (cell counting at day 2, n = 2 independent experiments, mean percentage ± SEM). Insets are showing nuclei staining with Hoechst; (**C**) Relative expression of axial mesoderm (*T*, *FOXA2*, *SHH*, *FOXJ1*, *NOGGIN* and *NOTO*), endoderm (*GSC, CER1* and *SOX17*) and mesoderm (*MIXL1*, *TBX6* and *FOXF1*) markers expression (n = 4, mean values). ND = Non-Detected Ct value. Mean and standard error of mean (SEM) values relative to experiments in panel C are shown in Appendix B. Statistical analysis (2 way Anova test) relative to experiments in panel C to determine significant differences between conditions at day 2 and day 5 is shown in Appendix C. Scale bars: 100 µm.

**Figure 5 cells-09-00509-f005:**
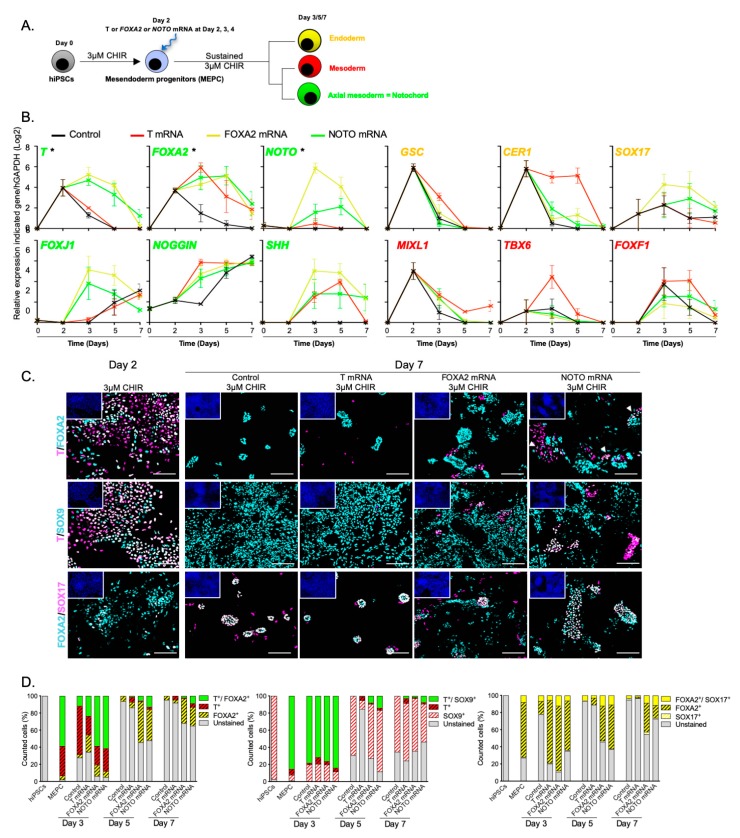
Generation and differentiation of mesendoderm progenitors upon WNT signalling activation following *T*, *FOXA2* or *NOTO* mRNA transfections. (**A**) Differentiation of MEPC following *T, FOXA2* or *NOTO* mRNA transfections; (**B**) Relative expression of axial mesoderm (*T*, *FOXA2 NOTO*, *SHH*, *NOGGIN* and *FOXJ1*), endoderm (*GSC*, *CER1* and *SOX17*) and mesoderm (*MIXL1*, *TBX6* and *FOXF1*) markers, (n = 3 independent experiments, mean values ± SEM). * indicates endogenous expression analysed by 3′UTR amplification of *T*, *FOXA2* and *NOTO transcripts*; (**C**,**D**) Immunostainings and quantifications of T+/FOXA2+, T+/SOX9+ and FOXA2+/SOX17+ cells (n = 2; quantification n = 2 technical replicates, mean values). Insets in C are showing nuclei staining with Hoechst. Scale bars: 100 µm.

**Figure 6 cells-09-00509-f006:**
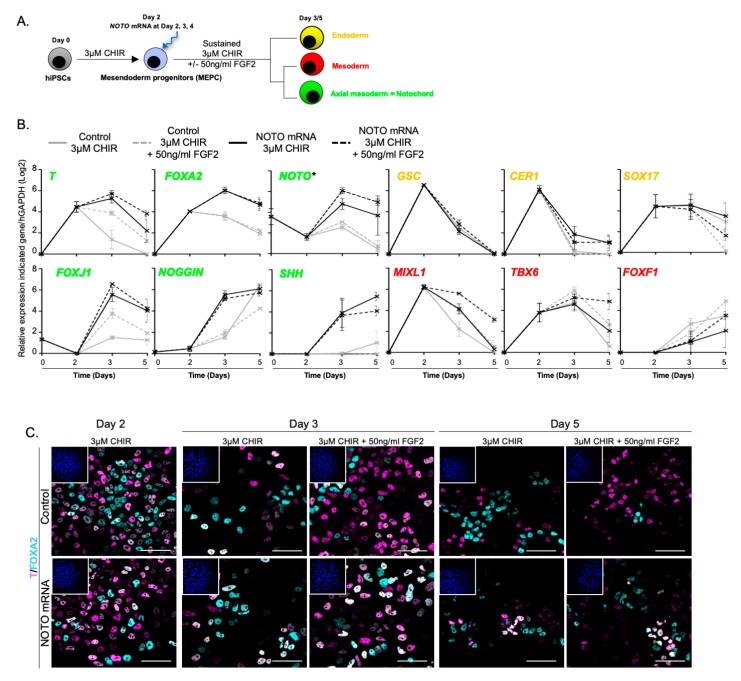
The FGF signalling pathway does not enhance notochordal differentiation. (**A**) Assessment of MEPC differentiation into NLC following FGF2 supplementation; (**B**) Relative expression of axial mesoderm (*T*, *FOXA2*, *NOTO*, *FOXJ1*, *NOGGIN* and *SHH*), endoderm (*GSC*, *CER1* and *SOX17*) and mesoderm (*MIXL1*, *TBX6* and *FOXF1*) markers in differentiating MEPC (RT-qPCR, n = 2 independent experiments, mean values ± SEM). * endogenous expression analysed by 3′UTR amplification of *NOTO transcript*; (**C**) Immunostaining of T+/FOXA2+ positive cells in differentiating MEPC (n = 2). Insets are showing nuclei stained with Hoechst. Scale bar: 50 μm.

**Figure 7 cells-09-00509-f007:**
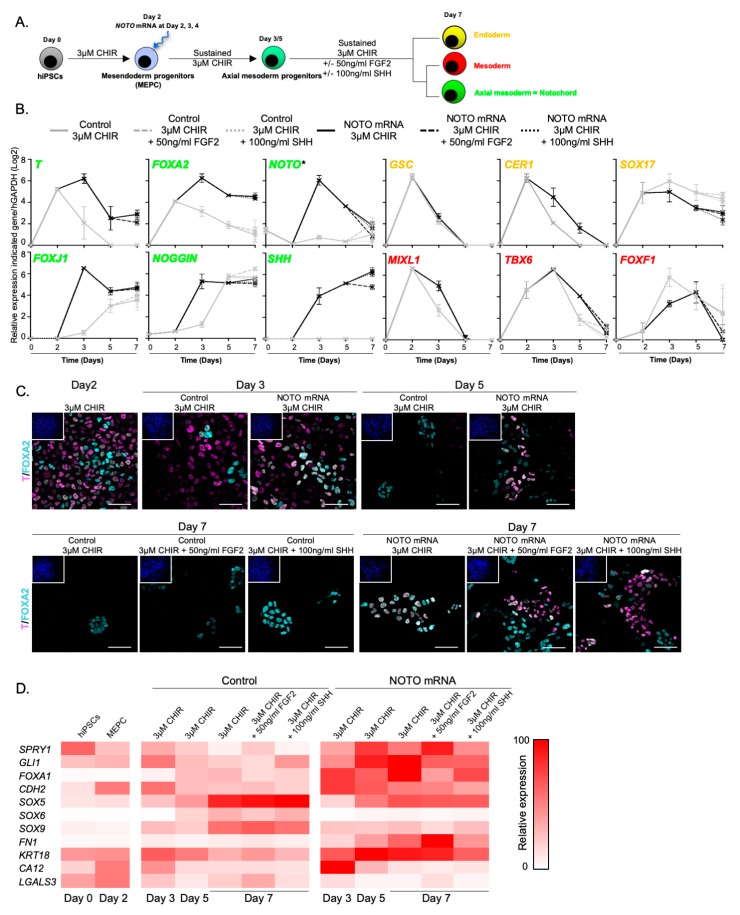
*NOTO* mRNA transfection and WNT signalling activity are sufficient to induce a stable NLC population. (**A**) Assessment of NLC stabilization by FGF and SHH signalling activities; (**B**) Relative expression of axial mesoderm (*T*, *FOXA2*, *NOTO*, *SHH*, *FOXJ1*, *NOGGIN* and *SHH*), endoderm (*GSC*, *CER1*, *SOX17*) and mesoderm (*MIXL1*, *TBX6* and *FOXF1*) markers (n = 2 independent experiments, mean ± SEM). * Endogenous expression analysed by 3′UTR amplification of *NOTO transcript*; (**C**) Immunostainings of T+/FOXA2+ positive cells. Insets are showing nuclei staining with Hoechst. Scale bars: 50 µm; (**D**) Heatmap representation of gene expression profiles during control and *NOTO*-transfected cell differentiation (n = 2, mean values).

**Figure 8 cells-09-00509-f008:**
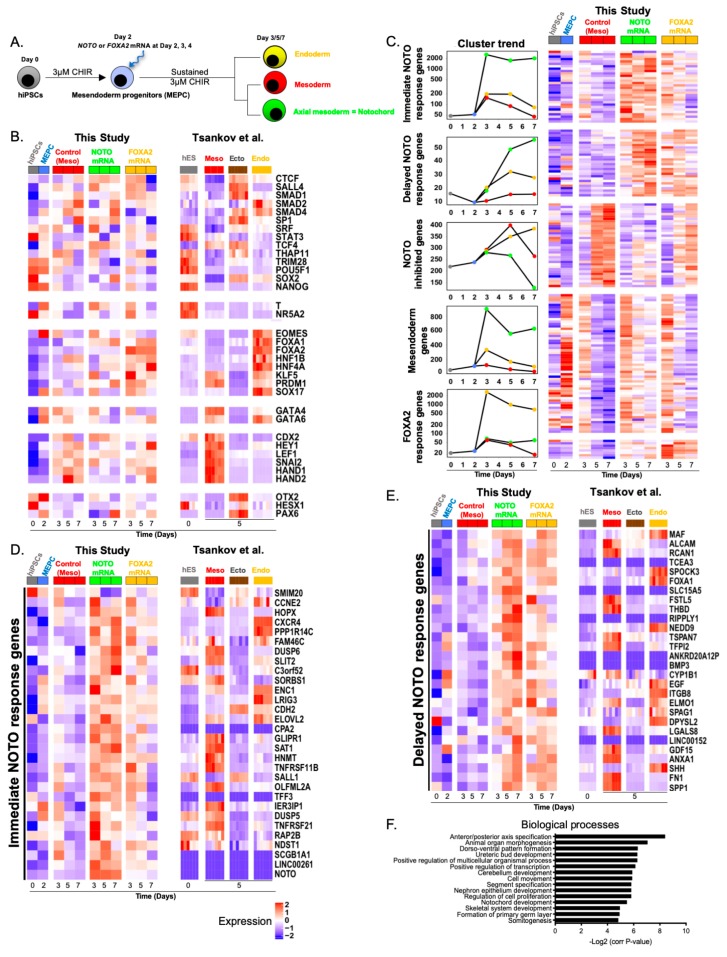
*NOTO* mRNA transfection induced a distinct molecular signature. (**A**) Differentiation of MEPC following *NOTO* or *FOXA2* mRNA transfections; (**B**) Expression levels of genes used as markers of mesoderm, ectoderm and endoderm across our samples (left) or Tsankov et al. samples (right); (**C**) RNAseq expression profile of differentially expressed genes during the course of differentiation. Differentially expressed genes were distributed in 5 clusters based on their kinetic of expression; (**D**) Expression levels of immediate *NOTO* response genes during the course of *NOTO*- and *FOXA2*-driven MEPC differentiation (this study) and in hESC-derived mesoderm, ectoderm and endoderm [59]; (**E**) Expression levels of delayed *NOTO* response genes during the course of *NOTO*- and *FOXA2*-driven MEPC differentiation (this study) and in hESC-derived mesoderm, ectoderm and endoderm [59]; (**F**) Top 15 Biological Processes associated with the up-regulated genes in *NOTO*-transfected condition compared to *FOXA2*-transfected condition. Cluster details for mesendoderm genes, NOTO inhibited genes and FOXA2 response genes are presented in Appendix A.

**Figure 9 cells-09-00509-f009:**
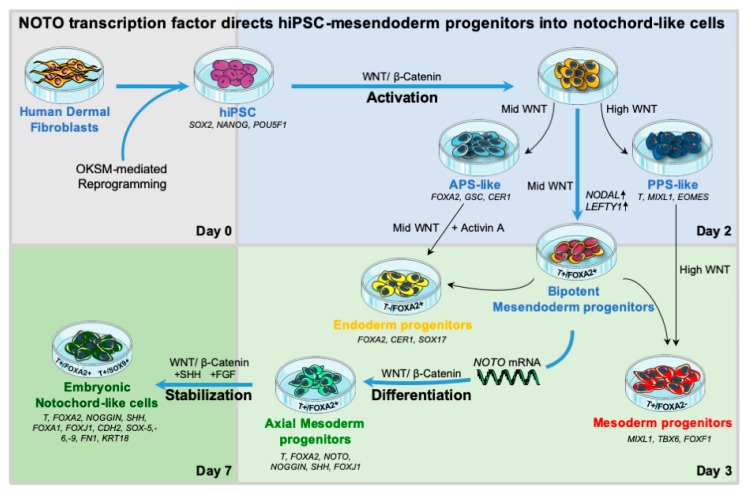
Visual summary of the main findings of the study. Human iPSCs were generated from dermal fibroblasts using 4 reprogramming factors (Oct4, Klf4, Sox2 and c-Myc = OKSM). Activation at day 2: WNT/β-catenin pathway activity (intermediate concentration of CHIR = Mid WNT) induced an increase in *NODAL* and *LEFTY1* gene expression. In this condition, high levels of bipotent mesendoderm progenitors (T+/FOXA2+ cells) were generated. Supplementation with Activin A resulted in hiPSCs commitment toward endoderm lineage (T-/FOXA2+ progenitor cells). High WNT pathway activation (high concentration of CHIR = High WNT) resulted in hiPSCs commitment toward mesoderm lineage (T+/FOXA2- progenitor cells). Differentiation from day 3: Mesendoderm progenitors transfected with synthetic mRNAs encoding human NOTO transcription factor and sustained with Mid WNT signalling activation generated axial mesoderm progenitors (T+/FOXA2+ cells). Stabilization up to day 7: *NOTO* transfection and Mid WNT signalling activation increased both SHH and FGF signalling pathway activities in axial mesoderm progenitors, which further differentiated into stable notochord-like cell population (NLC) at day 7 (T+/FOXA2+ cells and T+/SOX9+ cells). Human iPSC-derived NLC expressed embryonic notochord-related markers. Blue arrows indicate optimal culture condition for notochordal differentiation. Sets of gene markers relative to lineages or specific cell-types are indicated.

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
