# Peer review of "NOTO Transcription Factor Directs Human Induced Pluripotent Stem Cell-Derived Mesendoderm Progenitors to a Notochordal Fate"

_cells, 2020, doi:10.3390/cells9020509_

Round 1

Reviewer 1 Report

This is an interesting story , and the content is too complicated to be understood. I presume that the authors were trying to put as many things as they can. The introduction part should be rewritten to help your readers to know more regarding the background story of this work. If NOTO is the main transcription factor directing hIPS to a notochoral development, do you think that any inhibition effects from SiRNA, or shRNA to impair the function of the factor may show a negative results in your story?  I suggest the authors should set up a chart to illurstrate the relationships among these genes to help your readers have an integral concept on your findings. 

Reviewer 2 Report

The manuscript by Colombier et al. reports on a very detailed procedure to obtain human iPSC-derived mesendoderm progenitors to a notochord fate. The authors, interested in developing regenerative medicine strategies for intervertebral disc defects, have undertaken a deep characterization of hiPSC commitment towards notochord-like cells which are involved in disc homeostasis. This manuscript is well written, the messages delivered are clear and the data increase knowledge in stem cell field dedicated to improve procedures of hiPSC differentiation towards particular specific lineages.

Despite all these positive aspects this manuscript could still be ameliorate following the points raised below :

Major points:

Figures 2B, 3B and 4B: Relative expression levels of 2 independent experiments are shown which, indeed could not lead to statistical analysis. Since there are many samples with clear cut differential expression levels, the shown mean values are acceptable. However, for experiment shown in Figure 4B, the mean +/-Standard Deviation should be included and a statistical analysis could be provided since 4 samples have been processed.

Figure 2C and 3C: The pictures of the control cell line, without treatment, should be included.

Figures 2D, 3D, 4B and 5C: Merge pictures with Dapi and specific antibodies (not, as shown, in an inset) will allow to better visualize the proportion of positive cells for each antibody. Also higher magnification should be better for these figures.

Figure 5B: The expression of endogenous Noto is stimulated by transfection with Noto mRNA. This should be discussed.

Figure 7D: It should be informative to add some neuroectoderm and endoderm markers to prove the specificity of Noto mRNA towards mesoderm lineage.

The CHIR chemical is a well known inducer of mesoderm used towards cardiomyocyte lineage (for exemple). This should be mentioned. Did the authors observe cardiomyocytes in their culture along the differentiation procedure ? This should be stated.

Figure S2: This figure is supposed to be the details of transcriptomic clusters of Figure 8 (not Figure 4 as written). The “noto inhibited genes” cluster is well detailed and corresponds to the cluster shown in Figure 8 but for the other clusters, informations are missing. I did not identify to which cluster of Figure 8 those shown in S2 correspond. Please, be more precise.

The expression level of sox5, sox6 and sox9 should be searched and added in the heatmap of Figures 8B and 8C, since these genes are required for proper notochord development and maintenance, at least in the murine model. It should be very informative to add them.

Minor point

Ref. N°55 contains typing error.

Round 2

Reviewer 1 Report

I am happy to the revised version. 

Author Response

see pdf file

Reviewer 2 Report

The authors provided an accurate and convincing revised version of their manuscript.

All the raised concerns have been well answered.

However, it remains few minor points to be corrected:

Line 24: favour and not favours

Lines 262/263: part of the sentence is missing.

Line 435: Reference is not formatted.

Paragraphe 2.6 has not been updated with the new statistical analysis provided.

Lines 428 to 432: the flh mutants are not flh-/- mutants: it is not a Knock Out of the gene, the flh mRNA is still there. The authors have to correct the sentence accordingly with Reference 69:

FROM REF. 69: “However, flh RNA is expressed at lower levels in flh mutants suggesting that flh positively regulates its own expression”.

Also this result seems to contradict the following sentence:

“In the mouse model the loss of Noto function results in persistent Noto expression in anterior region of the embryo suggesting that Noto is required for its own repression [31].”

So it is not really clear whether NOTO induces or represses its own transcription.

This point should be better clarified.

Author Response

see PDF file
